

# The effect of walking speed on the foot inter-segment kinematics, ground reaction forces and lower limb joint moments

Dong Sun[1,2,3], Gusztáv Fekete[2,3], Qichang Mei[1,4] and Yaodong Gu[1,4]

[1] Faculty of Sports Science, Ningbo University, Ningbo, China
[2] Faculty of Engineering, University of Pannonia, Veszprem, Hungary
[3] Savaria Institute of Technology, Eötvös Lorand University, Szombathely, Hungary
[4] Auckland Bioengineering Institute, University of Auckland, Auckland, New Zealand

Corresponding author
Yaodong Gu,
guyaodong@nbu.edu.cn

## ABSTRACT

**Background:** Normative foot kinematic and kinetic data with different walking speeds will benefit rehabilitation programs and improving gait performance.
The purpose of this study was to analyze foot kinematics and kinetics differences between slow walking (SW), normal walking (NW) and fast walking (FW) of healthy subjects.

**Methods:** A total of 10 healthy male subjects participated in this study; they were asked to carry out walks at a self-selected speed. After measuring and averaging the results of NW, the subjects were asked to perform a 25% slower and 25% faster walk, respectively. Temporal-spatial parameters, kinematics of the tibia (TB), hindfoot (HF), forefoot (FF) and hallux (HX), and ground reaction forces (GRFs) were recorded while the subjects walked at averaged speeds of 1.01 m/s (SW), 1.34 m/s (NW), and 1.68 m/s (FW).

**Results:** Hindfoot relative to tibia (HF/TB) and forefoot relative to hindfoot (FF/HF) dorsiflexion (DF) increased in FW, while hallux relative to forefoot (HX/FF) DF decreased. Increased peak eversion (EV) and peak external rotation (ER) in HF/TB were observed in FW with decreased peak supination (SP) in FF/HF. GRFs were increased significantly with walking speed. The peak values of the knee and ankle moments in the sagittal and frontal planes significantly increased during FW compared with SW and NW.

**Discussion:** Limited HF/TB and FF/HF motion of SW was likely compensated for increased HX/FF DF. Although small angle variation in HF/TB EV and FF/HF SP during FW may have profound effects for foot kinetics. Higher HF/TB ER contributed to the FF push-off the ground while the center of mass (COM) progresses forward in FW, therefore accompanied by higher FF/HF abduction in FW. Increased peak vertical GRF in FW may affected by decreased stance duration time, the biomechanical mechanism maybe the change in vertical COM height and increase leg stiffness. Walking speed changes accompanied with modulated sagittal plane ankle moments to alter the braking GRF during loading response. The findings of foot kinematics, GRFs, and lower limb joint moments among healthy males may set a reference to distinguish abnormal and pathological gait patterns.

## INTRODUCTION

Walking speed is a parameter that is interconnected with the measures of mobility and daily activities. It is also worth mentioning that the ability of changing walking speed is a personal measure of adaptation with regard to the lower limbs which can be quantified by various gait parameters. Significant relevance have been found between walking speed and spatial-temporal parameters, lower limb kinematics, kinetics, and electromyography activities in both healthy and pathology adults (*Andriacchi, Ogle & Galante, 1977*; *Chiu & Wang, 2007*; *Chung & Wang, 2010*; *Dubbeldam et al., 2010*; *Holden, Chou & Stanhope, 1997*; *Tulchin et al., 2009*; *Van Hoeve et al., 2017*; *Fukuchi, Fukuchi & Duarte, 2018*).

It has been also demonstrated that an initial increase in stride length is followed by increased cadence together with increased walking speed in healthy adults (*Kirtley, 2006*). Changing walking speed has been found to affect sagittal plane kinematics as well. Increased peak knee flexion and ankle dorsiflexion (DF) angles during stance phase with a higher walking velocity was reported (*Umberger & Martin, 2007*), while other authors proved that significant differences were found at the hindfoot (HF) and forefoot (FF) sagittal kinematics in healthy adults with different walking speeds (*Tulchin et al., 2009*). *Dubbeldam et al. (2010)* found that walking speed significantly affected foot and ankle kinematics, which showed that minimum tibio-talar plantarflexion (PF) and maximum hallux (HX) DF decreased during the toe-off (TO) phase as walking speed decreased. *Grant & Chester (2015)* found all significant changes in the magnitude of foot angles during variable walking speed only occurred in the sagittal plane. Non-sagittal foot kinematics of adult subjects were also reported in recent studies. However, with an increase in walking speed, the HF was found to be more inverted on the frontal plane and more adducted on the transverse plane with respect to the tibia (TB) during late stance (*Caravaggi, Leardini & Crompton, 2010*). Gait kinetic parameters showed a strong correlation with an increase in walking speed, particularly the ground reaction forces (GRFs) and joint moments. It was reported that vertical and horizontal GRFs were significantly affected by gait speed, especially during the loading response phase and mid-stance phase (*White et al., 1996*). A linear correlation between peak propulsive force and joint moments and walking speed was reported (*Lelas et al., 2003*). The addition of intra-foot kinetics to multi-segment foot modeling may provide additional insight into foot and ankle function under different walking speed. A three-segment kinetic foot model was mentioned in a previous study, which incorporating kinetic parameters and calculating joint moments and powers of ankle, mid-tarsal, and first metatarsophalangeal joints (*Bruening, Cooney & Buczek, 2012*). While *Dixon, Böhm & Döderlein (2012)* attempt to use a four segments Oxford foot model (OFM) to calculate ankle and midfoot kinetics and compare ankle kinetics with one foot segment Plug-in-Gait (PIG) model, the masses of the FF and HF were set to half the value of the whole foot. Their study mainly suggested the overestimation of ankle power when using the one-segment food model.

The multi-segment foot models have been extensively used by dividing the foot into multiple rigid segments over the past decades (*Caravaggi et al., 2010*;

*MacWilliams, Cowley & Nicholson, 2003*). The OFM has demonstrated robust validity and repeatability with an overall foot inter-segment angles throughout a gait cycle in both adults and children (*Carson et al., 2001*). Numerous studies have examined the effects of gait speed on temporal-spatial parameters, joint kinematics, GRFs, and muscle activities (*Chiu & Wang, 2007*; *Goldberg & Stanhope, 2013*; *Kang & Dingwell, 2008*). However, the effect of walking speed remained controversial on foot kinematics.

A validated reference set, with regard to the inter-segmental kinematics of normal foot, could be a substantial result in order to compare abnormal or pathological gait to clinical gait. Therefore, the main objective of this study was to investigate differences in foot inter-segment kinematics, GRFs, and lower limb joint moment variables between self-selected walking speed (NW), relatively slower walking speed (SW) and relatively faster walking speed (FW) in matured healthy subjects. We hypothesized that with increasing walking speed, foot inter-segment kinematics and range of motion (ROM) would significantly increase in the sagittal, frontal, and transverse planes. For kinetic variables, increased GRFs, and joint moments were also hypothesized with increasing walking speed.

## METHODS

### Subjects

A total of 10 healthy male subjects (age: 24.6 ± 2.3 years, height: 1.72 ± 0.3 m, weight: 67.6 ± 8.3 kg, BMI: 22.85 ± 1.22 kg/m$^2$) without misalignment of lower limbs and any orthopedic surgeries history of lower extremities volunteered to participate in this study. All the subjects were rear-foot strikers and the dominant feet are the right one based on the test of kicking a soccer ball, so all statistics concerning the tests are based on data from the right feet. Informed consent were obtained before taking part in the experiments. Ethics approval was obtained from Ethics committee of Ningbo University (No. RAGH20170516).

### Experimental protocol and procedure

The three-dimensional marker trajectories were captured using a VICON MX motion analysis system (Oxford Metrics Ltd, Oxford, UK), with eight cameras and sampling at 200 Hz. Reflective markers were placed according to the marker set of a modified version of the OFM (*Stebbins et al., 2006*). The OFM is a multi-segment foot model used to calculate intersegment foot kinematics. Simplified complex foot structure to three rigid segments (TB, HF, and FF) and one vector (HX). A set of 36 markers (nine mm) were placed bilaterally on bony landmarks to model the TB, HF, FF, and HX. Static marker placement on the right foot was shown in Fig. 1. All inter-segment motions except HX motion were three-dimensional. The kinematic sign conventions used for OFM inter-segment angles were shown in Table 1. The OFM have been shown to provide consistent angular motion throughout the gait cycle for adults and children. However, marker placement variations may have potential influence on joint angles assessment. For each subject, the right foot were measured in three sequential walking speed sessions, between which the markers were not removed. In order to prevent high variability between

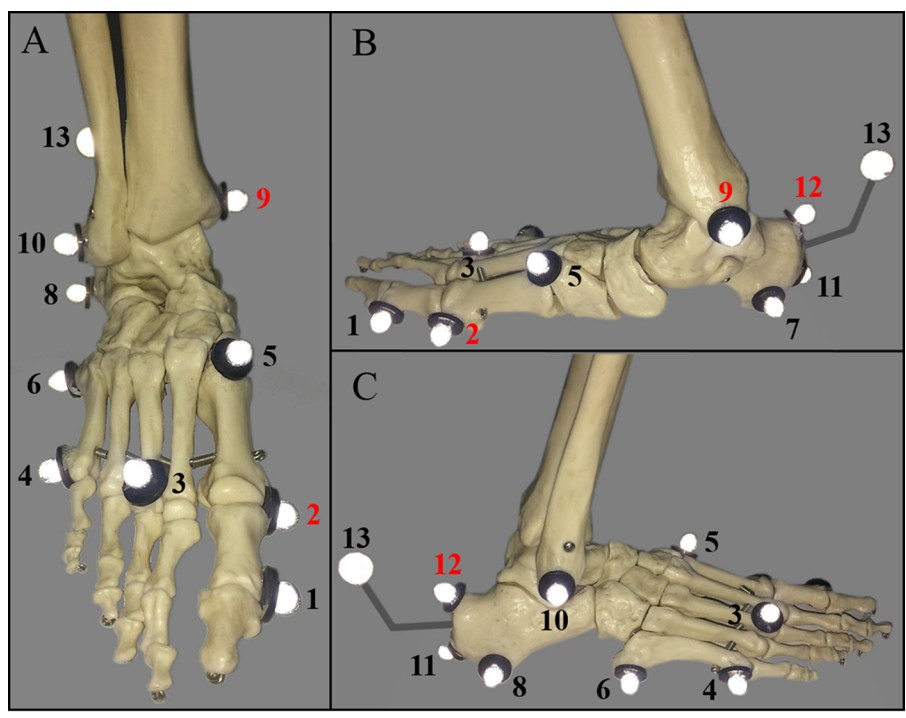

| 1: RHLX | 4: RD5M | 7: RSTL | 10: RANK | 13: RCPG |
|---------|---------|---------|----------|----------|
| 2: RD1M | 5: RP1M | 8: RLCA | 11: RHEE | |
| 3: RTOE | 6: RP5M | 9: RMMA | 12: RPCA | |

**Figure 1 Oxford foot model marker placement.** (A) the dorsal view; (B) the medial view; and (C) the lateral view. The three markers (RD1M, RMMA and RPCA) were removed after static modeling.

**Table 1 Kinematic sign conventions used for OFM inter-segment angles.**

| Foot segment angles | Sagittal | Frontal | Transverse |
|---------------------|----------|---------|------------|
| Hindfoot/tibia | Dorsiflexion (DF): + | Inversion (IV): + | Internal rotation (IR): + |
| | Plantarflexion (PF): − | Eversion (EV): − | External rotation (ER): − |
| Forefoot/hindfoot | Dorsiflexion (DF): + | Supination (SP): + | Adduction (ADD): + |
| | Plantarflexion (PF): − | Pronation (PR): − | Abduction (ABD): − |
| Forefoot/tibia | Dorsiflexion (DF): + | Supination (SP): + | Adduction (ADD): + |
| | Plantarflexion (PF): − | Pronation (PR): − | Abduction (ABD): − |
| Hallux/forefoot* | Dorsiflexion (DF): + | \ | \ |
| | Plantarflexion (PF): − | \ | \ |

**Note:**
* Sagittal plane only.

subjects, only one experienced laboratory technician were asked to attach the markers to all participants. Calibration of the system was performed before the gait trials. GRFs data were collected using a 600 × 400 mm force platform (AMTI, Watertown, MA, USA) at a sampling frequency of 1,000 Hz. The marker and force data were synchronized.

Initially, the subjects were required to walk at their own comfortable walking speed (normal walking, NW). The walking speed was recorded using Brower timing lights

(Brower Timing System, Draper, UT, USA), positioned 1.2 m apart, centering the force plate. In the second and third session the subjects were asked to walk 25% slower than NW (slow walking, SW) and 25% faster than NW (fast walking, FW) of their comfortable walking speed (NW), respectively. During each session, eight successful trials were obtained for each walking speed with less than 5% variance and within 5% of the predefined walking speed.

## Data analyses

Temporal-spatial parameters, including walking speed, stride length, stride time, stance duration and cadence, were calculated and recorded. All gait data were labelled and processed in VICON Nexus software (Version 1.8.5, VICON). Euler angles were calculated for inter-segment rotation following the sequence of Grood and Suntay (flexion, adduction (ADD), and rotation) (*Grood & Suntay, 1983*). The following three-dimensional motion of angles were measured: hindfoot relative to tibia (HF/TB), forefoot relative to hindfoot (FF/HF), forefoot relative to tibia (FF/TB), and hallux relative to forefoot (HX/FF) (sagittal plane only). Kinematic and kinetic traces were visually checked and inconsistent trials removed. Eight successful measurements were recorded when subjects landed with right foot on the force platform. For each trial, gait events were detected by the use of the vertical GRF to determine initial heel contact (HC) and TO with a threshold of 30N. OFM parameters such as the angular motion at HC, TO, peak values, and ROM in the stance phase were derived with regard to the different walking speeds. Peak propulsive, braking, and vertical GRFs were also recorded and analyzed. Vertical average loading rate (VALR) was computed as first peak vertical loading GRF/time (from contact). The peak braking GRFs and peak propulsive GRFs were the peak negative and positive anterior-posterior GRFs.

Oxford foot model does not calculate moments, it is a kinematics-only model (*Paterson et al., 2017*). Therefore, the knee and ankle external moment data in the sagittal and frontal planes in this study were calculated using a standard full inverse dynamic model based on the VICON PIG model (*Keenan et al., 2011*). GRFs and joint moments were normalized and reported in body weight (BW) and Nm/kg. The foot kinematic and GRF data were filtered with a fourth-order zero-phase lag Butterworth filter having a cutoff frequency of eight Hz and 20 Hz, respectively. One gait cycle was defined as ipsilateral HC ground twice. Gait cycles were normalized in the complete ROM in order to clearly distinguish major and minor variations in the patterns of the individual trials. Spatiotemporal parameters, foot kinematics, GRFs, and joint moments were determined in case of each participant during the three different walking speeds. The collected foot inter-segment kinematics for analysis include HC, TO, and peak angle values in a stance phase and angle curves during a gait cycle. The ROM was defined as the maximum minus minimum angle during stance phase.

## Statistical analyses

The statistical analyses were performed in SPSS version 22.0 (SPSS Science, Chicago, IL, USA). All data were normally distributed which assessed by the Shapiro–Wilk test and

**Table 2 Descriptive statistics of the temporal-spatial parameters (mean ± SD) during three walking speeds.**

| Variables | SW | NW | FW |
|---|---|---|---|
| Walking speed** (m/s) | 1.01 ± 0.06 | 1.34 ± 0.05 | 1.69 ± 0.12 |
| Stride time** (s) | 1.11 ± 0.02 | 0.96 ± 0.02 | 0.79 ± 0.02 |
| Stance duration** (s) | 0.65 ± 0.02 | 0.55 ± 0.02 | 0.39 ± 0.02 |
| Stance phase* (% gait cycle) | 57.99 ± 7.81 | 56.31 ± 6.54 | 48.27 ± 2.84 |
| Cadence** (strides/min) | 108.6 ± 3.9 | 126.5 ± 2.9 | 154.3 ± 4.5 |
| Stride length* (m) | 1.31 ± 0.04 | 1.37 ± 0.04 | 1.42 ± 0.06 |

Notes:
  * Statistical significance $p < 0.01$.
  ** Statistical significance $p < 0.001$.

are presented as mean ± SD (standard deviation). One-way analysis of variance (ANOVA) was performed, followed by a post hoc Bonferroni test, were used to test for significant differences in temporal-spatial parameters, foot inter-segment kinematics, GRFs, and lower limb joint moments with changes of walking speed. Statistical significance was determined at the level of $p < 0.05$.

# RESULTS

## Kinematics

### Temporal-spatial parameters

Mean and SD values of the temporal-spatial variables of SW, NW, and FW were given in Table 2. The one-way ANOVA test proved that the temporal variables are statistically significant with regard to the three walking speeds. With increasing walking speed, the stride length, and the cadence increased while the stance duration, stance phase (% gait cycle) and stride time decreased.

### Hindfoot relative to tibia motion

Average foot inter-segment kinematics curves during gait cycle are shown in Fig. 2. Selected crucial points and statistics data about foot motions during stance phase, values at HC, TO, maximum (max) and minimum (min) foot angles, ROM are included in Table 3. No significant differences were observed in the HF/TB motion between SW and NW in sagittal, frontal or transverse plane. Nevertheless, one-way ANOVA showed a statistically significant difference between the movements. In sagittal plane, FW showed a significantly higher HC, DF, peak DF, and DF/PF ROM. An increase in walking speed were corresponded with a significant increase of the peak values of HF/TB DF. In frontal plane, a significant increase in eversion (EV) at HC was found at higher walking speed during stance phase, while peak inversion angle was found lower in FW inversely. In transverse plane, significantly greater peak external rotation (ER) and ROM were found in FW.

### Forefoot relative to hindfoot motion

In the sagittal plane, peak FF/HF DF angle in FW was significantly greater than SW. Peak PF angle was found to be significantly higher in NW. Concerning ROM values,
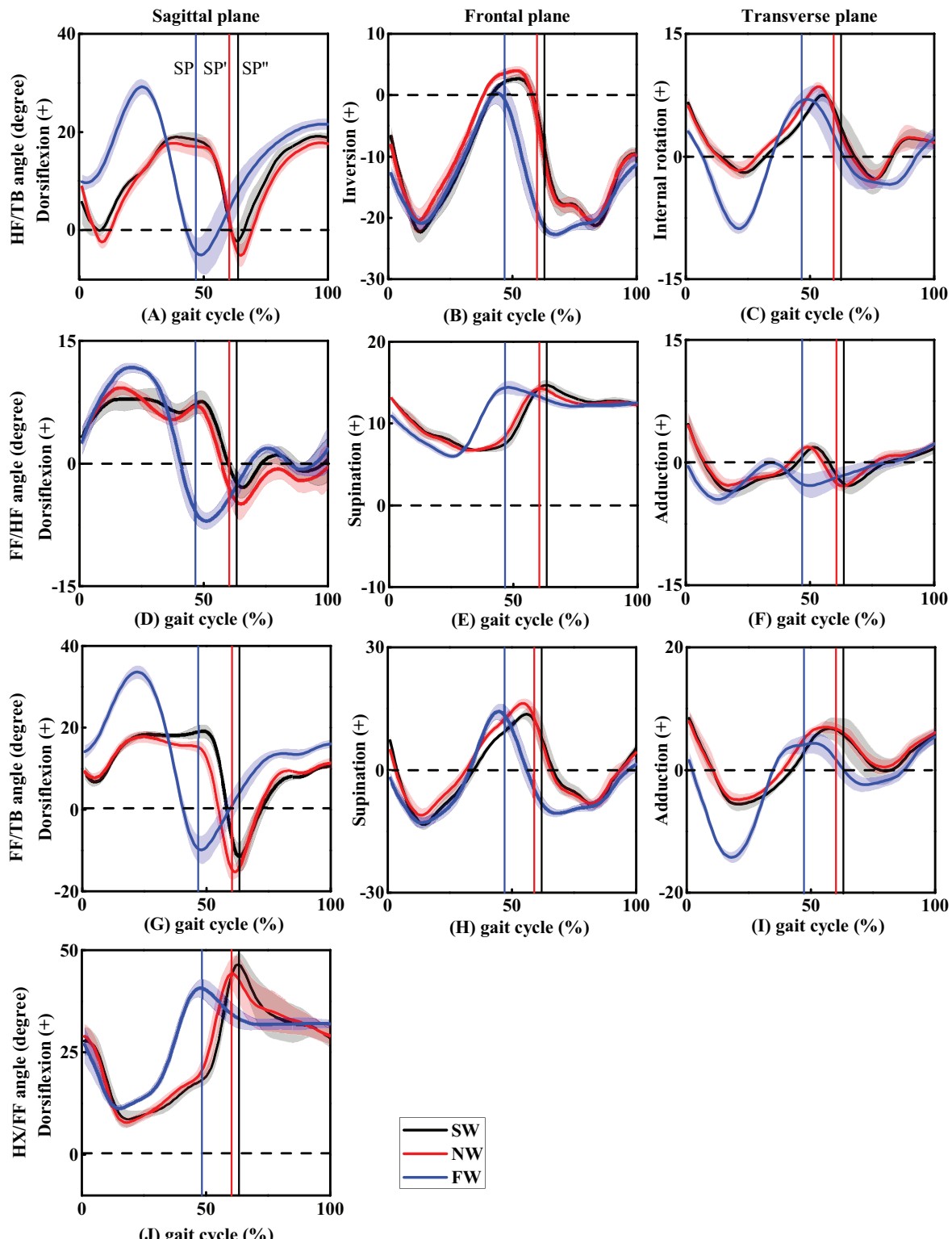

**Figure 2 The trend of foot inter-segment kinematics (°) in sagittal, frontal and transverse planes during one gait cycle (A–C, HF/TB; D–F, FF/HF; G–I, FF/TB; J, HX/FF).** Solid line with shade represents mean ± standard deviation across three walking speeds. SP, SP′, and SP″ represents the stance phase of FW, NW, and SW conditions.

**Table 3 Foot kinematics with three walking speeds in the stance phase of gait.**

| Variables | Walking speed | | | ANOVA p-value | Post hoc test (p-value) | | |
|---|---|---|---|---|---|---|---|
| | SW | NW | FW | | SW/NW | SW/FW | NW/FW |
| **Hindfoot/tibia (HF/TB)** | | | | | | | |
| DF (IC) | 5.7 ± 1.6 | 6.0 ± 2.2 | 9.7 ± 1.3 | <0.001 | 0.17 | <0.001 | <0.001 |
| DF (max) | 19.1 ± 1.2 | 17.7 ± 0.9 | 29.5 ± 1.5 | <0.001 | 0.06 | <0.001 | <0.001 |
| PF (max) | 0.1 ± 3.0 | −0.3 ± 1.6 | −0.6 ± 2.2 | 0.33 | 0.16 | 0.53 | 0.84 |
| ROM (DF/PF) | 21.0 ± 1.4 | 22.3 ± 1.7 | 30.1 ± 2.9 | <0.001 | 0.17 | <0.001 | <0.001 |
| EV (IC) | −9.0 ± 1.6 | −10.1 ± 2.0 | −13.8 ± 1.3 | <0.001 | 0.43 | <0.001 | <0.001 |
| EV (max) | −22.6 ± 1.6 | −21.2 ± 2.1 | −21.5 ± 1.2 | 0.29 | 0.12 | 0.31 | 0.53 |
| IV (max) | 3.2 ± 1.8 | 3.9 ± 0.8 | −0.3 ± 1.5 | <0.001 | 0.31 | <0.001 | <0.001 |
| ROM (IV/EV) | 25.8 ± 1.64 | 25.0 ± 2.4 | 21.2 ± 2.7 | <0.001 | 0.55 | <0.001 | <0.001 |
| IR (IC) | 2.7 ± 0.8 | 2.9 ± 1.1 | 2.6 ± 0.7 | 0.67 | 0.71 | 0.61 | 0.38 |
| ER (max) | −2.0 ± 0.6 | −2.3 ± 0.6 | −9.0 ± 0.7 | <0.001 | 0.70 | <0.001 | <0.001 |
| IR (max) | 7.2 ± 1.5 | 8.4 ± 0.5 | 5.6 ± 0.8 | 0.59 | 0.28 | 0.39 | 0.72 |
| ROM (IR/ER) | 9.2 ± 1.3 | 10.4 ± 1.1 | 14.6 ± 1.1 | <0.001 | 0.11 | <0.001 | <0.001 |
| **Forefoot/hindfoot (FF/HF)** | | | | | | | |
| DF (IC) | 2.8 ± 1.7 | 3.5 ± 1.0 | 3.8 ± 1.8 | 0.14 | 0.35 | 0.08 | 0.30 |
| DF (max) | 9.0 ± 0.7 | 9.2 ± 0.5 | 12.1 ± 1.2 | 0.01 | 0.23 | <0.001 | 0.02 |
| PF (max) | −9.4 ± 1.4 | −10.5 ± 0.9 | −5.6 ± 0.4 | <0.001 | 0.03 | <0.001 | <0.001 |
| ROM (DF/PF) | 18.3 ± 1.4 | 19.8 ± 0.9 | 14.2 ± 1.2 | <0.001 | 0.03 | <0.001 | <0.001 |
| SP (IC) | 12.2 ± 0.5 | 12.4 ± 0.4 | 10.7 ± 0.7 | <0.001 | 0.53 | <0.001 | <0.001 |
| SP (min) | 6.6 ± 0.3 | 6.5 ± 0.3 | 5.9 ± 0.3 | <0.001 | 0.98 | <0.001 | <0.001 |
| SP (max) | 14.6 ± 0.8 | 14.5 ± 0.6 | 13.8 ± 0.5 | 0.03 | 0.86 | 0.02 | 0.03 |
| ROM (SP/PR) | 8.0 ± 0.8 | 8.0 ± 0.5 | 7.9 ± 0.6 | 0.95 | 0.86 | 0.75 | 0.90 |
| ADD (IC) | 4.0 ± 1.6 | 3.7 ± 1.7 | −1.3 ± 0.9 | <0.001 | 0.72 | <0.001 | <0.001 |
| ABD (max) | −3.4 ± 0.9 | −2.8 ± 0.6 | −4.7 ± 0.6 | <0.001 | 0.14 | 0.01 | <0.001 |
| ABD (TO) | −1.6 ± 0.7 | −2.3 ± 0.6 | −1.6 ± 0.9 | 0.15 | 0.09 | 0.90 | 0.09 |
| ROM (ADD/ABD) | 7.4 ± 1.1 | 6.8 ± 1.2 | 5.1 ± 0.5 | <0.001 | 0.20 | <0.001 | 0.02 |
| **Forefoot/tibia (FF/TB)** | | | | | | | |
| DF (IC) | 8.6 ± 1.1 | 9.0 ± 1.1 | 14.4 ± 1.4 | <0.001 | 0.44 | <0.001 | <0.001 |
| DF (max) | 19.3 ± 1.3 | 17.8 ± 1.3 | 33.4 ± 1.8 | <0.001 | 0.04 | <0.001 | <0.001 |
| PF (max) | −7.0 ± 4.0 | −12.7 ± 2.2 | −4.8 ± 2.2 | <0.001 | <0.001 | 0.12 | <0.001 |
| ROM (DF/PF) | 26.3 ± 3.5 | 30.6 ± 2.8 | 38.2 ± 3.2 | <0.001 | 0.01 | <0.001 | <0.001 |
| SP (IC) | 3.2 ± 1.5 | 2.2 ± 1.9 | −3.3 ± 1.5 | <0.001 | 0.23 | <0.001 | <0.001 |
| PR (max) | −13.7 ± 1.9 | −12.2 ± 2.2 | −12.8 ± 1.4 | 0.22 | 0.09 | 0.29 | 0.45 |
| SP (max) | 15.6 ± 2.9 | 17.5 ± 1.2 | 13.9 ± 1.8 | 0.01 | 0.07 | 0.03 | 0.02 |
| ROM (SP/PR) | 29.3 ± 2.9 | 29.7 ± 1.9 | 26.8 ± 1.8 | 0.12 | 0.73 | 0.15 | 0.09 |
| ADD (IC) | 6.2 ± 1.5 | 6.0 ± 1.9 | 0.4 ± 1.3 | <0.001 | 0.84 | <0.001 | <0.001 |
| ABD (max) | −5.6 ± 1.1 | −4.8 ± 0.9 | −14.2 ± 0.9 | <0.001 | 0.11 | <0.001 | <0.001 |
| ADD (TO) | 7.5 ± 1.0 | 7.8 ± 1.0 | 4.1 ± 1.3 | <0.001 | 0.51 | <0.001 | <0.001 |
| ROM (ADD/ABD) | 13.1 ± 1.3 | 12.7 ± 0.7 | 18.2 ± 1.8 | <0.001 | 0.55 | <0.001 | <0.001 |

| Variables | Walking speed | | | ANOVA p-value | Post hoc test (p-value) | | |
|---|---|---|---|---|---|---|---|
| | SW | NW | FW | | SW/NW | SW/FW | NW/FW |
| **Hallux/forefoot (HX/FF)** | | | | | | | |
| DF (IC) | 27.3 ± 3.6 | 28.7 ± 3.0 | 25.5 ± 4.0 | 0.18 | 0.39 | 0.31 | 0.07 |
| DF (min) | 7.8 ± 1.7 | 7.5 ± 1.2 | 11.3 ± 0.7 | <0.001 | 0.98 | <0.001 | <0.001 |
| DF (max) | 46.1 ± 3.5 | 45.4 ± 2.7 | 40.0 ± 2.3 | <0.001 | 0.56 | <0.001 | <0.001 |
| ROM (DF/PF) | 37.5 ± 3.2 | 38.1 ± 2.5 | 26.6 ± 2.5 | <0.001 | 0.51 | <0.001 | <0.001 |

**Notes:**
Angular rotation of the hindfoot relative to the tibia (HF/TB), the forefoot relative to the hindfoot (FF/HF), the forefoot relative to the tibia (FF/TB) and the hallux relative to the forefoot (HX/FF).
SW, slow walking; NW, normal walking; FW, fast walking; DF, dorsiflexion; PF, plantarflexion; SP, supination; PR, pronation; ADD, adduction; ABD, abduction; IV, inversion; EV, eversion; IR, internal rotation; ER, external rotation; ROM, range of motion; IC, initial contact; TO, toe-off; max, maximum; min, minimum.

both SW and NW were shown to be significantly higher DF/PF ROM compared with FW. In the frontal plane, there was a significantly larger supination (SP) in SW and NW at HC. In the transverse plane, ADD angles at initial contact (IC) and ADD/abduction (ABD) ROM were found to be significantly higher in SW. However, maximum ABD angle was found to be significantly greater in FW.

### Forefoot relative to tibia motion
Forefoot relative to tibia motion showed similar variations compared with HF/TB motion. In the sagittal plane, FW showed more DF at IC and peak values compared with SW and NW. However, peak PF was found to be significantly higher in NW. ROM (DF/PF) increased as walking speed escalated. No significant difference were found in peak SP and SP/pronation ROM. In the transverse plane, ADD angle at IC and maximum ADD angle were significantly higher in SW and NW. Peak ABD and ADD/ABD ROM were also found to be significantly higher in FW.

### Hallux relative to forefoot motion
Only sagittal plane HX/FF motion were concluded in this study, where the HX was significantly less dorsiflexed in FW compared with SW and NW. Additionally, DF/PF ROM in SW and NW were significantly higher than FW.

## Kinetics
### Ground reaction forces
Ground reaction forces of each subjects were normalized to BW. Normalized GRFs and VALR were shown in Table 4. In the anterior–posterior direction, peak braking force together with peak propulsive force was significantly higher in FW compared with SW, and peak propulsive forces were significantly higher in FW compared to SW and NW. Peak vertical forces in FW condition were significantly higher than NW. VALR (BW/s) showed discriminatory effects between different walking speeds.

### Lower limb joint moments
With regard to the parameters of joint moments, walking speed had a significant effect on knee and ankle sagittal and frontal plane joint moments (Nm/kg). In the sagittal plane,

**Table 4 Comparison of GRFs and peak joint moments at three walking speed.**

| Variables | Walking speed | | | ANOVA p-value | Post hoc test (p-value) | | |
|---|---|---|---|---|---|---|---|
| | SW | NW | FW | | SW/NW | SW/FW | NW/FW |
| **Ground reaction forces** | | | | | | | |
| Peak braking (BW) | 0.21 ± 0.03 | 0.22 ± 0.02 | 0.26 ± 0.03 | 0.01 | 0.39 | 0.01 | 0.03 |
| Peak propulsive (BW) | 0.23 ± 0.01 | 0.24 ± 0.01 | 0.28 ± 0.03 | 0.01 | 0.54 | 0.01 | <0.001 |
| Peak vertical (BW) | 1.36 ± 0.07 | 1.42 ± 0.04 | 1.71 ± 0.05 | <0.001 | 0.28 | 0.02 | <0.001 |
| VALR (BW/s) | 9.99 ± 1.03 | 12.04 ± 0.97 | 18.56 ± 1.44 | <0.001 | 0.03 | <0.001 | <0.001 |
| **Joint moments (peak values) (Nm/kg)** | | | | | | | |
| Knee flexion | 0.25 ± 0.06 | 0.31 ± 0.09 | 0.45 ± 0.11 | <0.001 | 0.02 | <0.001 | <0.001 |
| Knee extension | 0.62 ± 0.13 | 0.65 ± 0.14 | 0.76 ± 0.13 | <0.001 | 0.26 | <0.001 | <0.001 |
| Knee adduction | 0.08 ± 0.05 | 0.08 ± 0.07 | 0.12 ± 0.06 | 0.01 | 0.62 | <0.001 | <0.001 |
| Ankle dorsiflexion | 0.21 ± 0.03 | 0.15 ± 0.02 | 0.18 ± 0.03 | 0.01 | 0.11 | <0.001 | 0.04 |
| Ankle plantarflexion | 0.37 ± 0.07 | 0.41 ± 0.08 | 0.51 ± 0.13 | <0.001 | 0.16 | <0.001 | <0.001 |
| Ankle eversion | 0.11 ± 0.01 | 0.11 ± 0.02 | 0.14 ± 0.02 | 0.01 | 0.35 | 0.02 | <0.001 |

**Note:**
BW, body weight; VALR, vertical average loading rate.

peak knee flexion moment in FW was significantly higher compared with SW and NW. For peak knee extension moment, FW was significantly higher compared with SW and NW. Peak ankle DF moment in early stance phase were shown to be significantly greater in FW compared with SW and NW. Peak ankle PF moment in late stance phase was also significantly greater in FW compared to SW and NW. In the frontal plane, statistically significant differences were also noted at the knee and ankle joints. Peak knee ADD and peak ankle EV moments were significantly greater in FW compared to SW and NW. With an increasing walking speed, knee, and ankle joint moments in the sagittal and frontal planes increased as well, which indicates a positive and possibly linear correlation between walking speeds and joint moments.

# DISCUSSION

The purpose of this study was to identify whether healthy adults have different foot inter-segment kinematics, GRFs and lower limb joint moments when walking at a comfortable, relatively slower, and faster walking speeds. This current study can provide valuable insight into how foot inter-segments interact with each other and how the GRFs and joint moments vary during different walking speed. The findings on one hand provide guidance for which parameters may improve gait performance, while on the other hand also provide a reference for clinical gait dysfunction diagnosis.

## Foot kinematics

Significant difference has been revealed on the relative segmental foot motions (HF/TB, FF/HF, FF/TB, and HX/FF) during SW, NW, and FW. It must be mentioned as well, that the relative segmental foot motions observed in SW, NW, and FW were nearly consistent with previous studies. Our findings showed that a higher DF angle was present at HC together with a higher maximum DF angle and sagittal plane ROM during FW with

regard to HF/TB. These results are also consistent with the previous findings (*Arnold et al., 2014*; *Grant & Chester, 2015*; *Jenkyn & Nicol, 2007*). These studies also suggest that walking in a higher speed indicates larger HF/TB DF angle. It was also concluded that in the HC HF/TB DF and peak HF/TB DF angles increased with walking speed. The increase in HF/TB DF enables a larger stride length, which could explain the temporal-spatial results in this study. Greater peak EV and frontal plane ROM of the HF/TB during the stance phase of FW were also found. Previous studies provided evidence that HF/TB peak EV could be used as an indicator for injuries risk due to excessive use (*Chang et al., 2014*; *Graham et al., 2011*). More ER of HF/TB was also found in FW during the mid-stance phase, which is likely to be credited to the increased step length and cadence. These parameters could be key factors to interpret higher HF/TB DF and ER.

Maximum FF/HF DF angles increased significantly in higher walking speed, while the angle variable was less than 3°. Furthermore, maximum FF/HF PF were significantly greater in FW, while the angle variable was less than 5°. This finding corresponds well with a previous study which also reported similar results in the FF/HF motion (*Tulchin et al., 2009*). Total sagittal FF/HF motion in plantar fasciitis subjects was found to be significantly greater compared to healthy subjects (*Chang et al., 2014*). One advantage of using the OFM was the ability to assess frontal and transverse plane foot motion. In this study, decreased maximum FF/HF SP angle was observed in FW (13.8° ± 0.5°) compared to SW (14.6° ± 0.8°) and NW (14.5° ± 0.6°). Variation trend of FF/HF SP angle at loading response was similar. Although differences in FF/HF motion between different walking speeds seem quite small at first glance, the stress and strain of the plantar fascia have shown a significant correlation with the intricate movements of the foot (*Caravaggi et al., 2010*). The plantar fascia tension would increase from 0.4 to 0.7 BW even when the arch angle changed 1° during the first 50% of stance phase (*Caravaggi, Leardini & Crompton, 2010*). In the transverse plane, FW (+25% NW) was shown to have more FF/HF ABD in mid-stance phase compared to SW (−20% NW) and NW. From earlier studies it was deduced that differences were found in FF motion in the transverse plane between different walking speeds, 50% NW, 75% NW and NW, respectively (*Dubbeldam et al., 2010*). More FF/HF ABD in 50% NW (−13.70° ± 5.64) was reported compared to 75% NW (−12.85° ± 5.58) and NW (−12.44° ± 5.74). FF/HF ABD angle showed similar variation in the speeds of 75% NW and NW. However, higher FF/HF ABD angle was also found in FW (125% NW) compared to 50% NW. During mid-stance phase to TO phase, the ER of HF contributed to the FF push-off over the ground while the center of mass (COM) progressed forward (*Papi, Rowe & Pomeroy, 2015*). For this reason, higher HF/TB ER angle may be accompanied by higher FF/HF ABD angle during walking with higher speed. The HX/FF was significantly more dorsiflexed in SW and NW throughout stance and additionally showed more ROM compared to FW. In our hypothesis, an increase in HX/FF DF angle was expected during walking with higher speed (FW), as a result of the higher loading, which would be transferred to the FF under higher walking speed. This is practically due to the roll-over mechanism of the foot (*Samson et al., 2014*). Nevertheless, lower HX/FF DF was observed during faster walking. A possible explanation is that the increased HX/FF DF angle may result in higher triceps

surae muscle activity and more tension on the plantar fascia, which may restrain FF and HX spreading in higher walking speed in this study (*Neptune, Kautz & Zajac, 2001*; *Neptune, Zajac & Kautz, 2004*). In our study, the foot inter-segment kinematic patterns between different walking speeds were similar to that measured by Van Hoeve (*Van Hoeve et al., 2017*) using OFM foot model. While some decreased ROM (FF/HF sagittal plane and HX/FF sagittal plane) during stance phase were found in faster walking speed (FW) in this study. *Caravaggi, Leardini & Crompton (2010)* found decreased ROM in the sagittal-plane between FF/HF with faster walking cadence, which were consistent FF/HF sagittal plane ROM between normal and faster walking speed. The decrease of FF/HF ROM was mainly caused by decreased maximal PF angle of FF. (*Tulchin et al., 2009*) found maximal plantarflexion showed a 4.5° change across walking speeds, as the duration of stance phase decreased. The decreased of HX/FF ROM during FW maybe a compensation mechanism for shorted stance time and decreasing FF/HF ROM.

In summary, in the sagittal plane, limited HF/TB, and FF/HF motions of SW were likely compensated by the increased HX/FF DF in mid-stance phase. In the frontal plane, increased peak EV angle in HF/TB was observed during FW, which was also accompanied with a decreased peak SP angle in FF/HF. The angle variation is low in the frontal plane, which has profound effects on foot morphology and on foot kinetics (*Ferber & Benson, 2011*; *Graham, Jawrani & Goel, 2011*). In the transverse plane, higher HF/TB ER contributed to the FF push-off the ground, while the COM progressed forward, therefore it was accompanied by higher FF/HF ABD in FW (*Wright et al., 2011*). The foot motion differences between SW, NW, and FW were consistent with the references which emphasizes the validity of the OFM for evaluating walking speed effects.

## Kinetics

In this study, kinetic parameters show a consistent correlation with increases in walking speed. Increased vertical GRF, braking GRF (the posterior component of the GRF vector), and propulsive GRF (the anterior component of the GRF vector) were found in FW. The vertical GRF spikes at loading response to form the first impact peak, and then increases at mid-stance phase to approximately 100–110% BW with a comfortable walking speed. Increased peak vertical GRF in FW was significantly affected by decreased stance duration time, which may minimize the change in vertical COM height, and in turn increase leg stiffness. Increased VALR was found during FW, peak vertical GRF, and VALR have been reported as risk factors of barefoot walking. *Zadpoor & Nikooyan (2006)* found an earlier rise in vertical GRF and greater VALR during barefoot walking, which were believed as risk factors for TB stress and plantar fasciitis. For normal adults walking, a change in the peak vertical GRF accompanied with a change in the horizontal component GRF (braking GRF and propulsive GRF). Braking and propulsive GRF increased with walking speeds from 1.0 to 2.0 m/s, and propulsive GRF generated during push-off phase could modulate walking speed (*Nilsson & Thorstensson, 1989*). Stride length and cadence could also influence propulsive GRF and increase with increasing walking speed, increases in peak propulsive GRF occur with increasing stride length at a higher walking speed (*Martin & Marsh, 1992*).

In the sagittal plane, peak knee, and ankle joint moments during stance were found to be significantly higher in FW compared to SW and NW. Lower limbs' peak joint moments were found systematically decreased with lower walking speed and could be well predicted based on walking speed among healthy individuals (*Lelas et al., 2003*). It was reported that peak sagittal plane moments had a predictive dependency on walking speed, and a correlation coefficient ($R = 0.86$) of linear regression with the peak external knee flexion moment was found in stance phase (*Schwartz, Rozumalski & Trost, 2008*). Significant correlation between lower limb joint moment and spatial-temporal parameters was also reported in a previous. Individuals only experienced a significant increase in lower limb joint moments with an increased stride length or both stride length and cadence (*Ardestani et al., 2016*). In this study, both cadence and stride length were found with increased walking speed, therefore greater joint moments were found in FW. Walking speed changes accompanied with modulated sagittal ankle and knee moments to alter the braking GRF during loading response. Moreover, increased sagittal ankle moment is a consequence of increased walking speed. As peak GRFs and joint moments generating more mechanical strain on soft tissues during FW, the ability of the individuals to regulate the GRFs and joint moments is essential to soft tissue health (*Dicharry, 2010*). Furthermore, it has been shown that a knee joint flexor moment can be caused by a larger plantarflexor moment to the ankle joint (*Simonsen et al., 1997*). In the frontal plane, the knee ADD moment significantly increased in FW Which was assessed as a marker of knee joint loading, the magnitude of knee ADD moment could reflect the medial compartment joint loading (*Hunt et al., 2006*). Further, due to the fact that lower limbs joints are interrelated and correlate by the kinematic and kinetic chain, alterations of foot kinematics may not only affect knee joint loads but may also have influence on the ankle joint (*Graham et al., 2011*). Kinetic parameters show a strong consistence with increases in walking speed, and was found mainly differences in FW compared to SW and NW.

## Limitations

Several limitations of this current study should be considered. First of all, human walk with shoes in the real life, the foot kinematic and joint kinetic data of this study was only collected during barefoot condition. Thus, the kinematic and kinetic patterns of shod conditions with different walking speed remains unknown. Secondly, the lowest walking speed selected in this study was about 1.0 m/s, which was considered not slow enough to represent or set reference for those patients with pathology on the foot, like diabetic foot or after stroke. Thirdly, intra-foot kinetics were not included in this study, the authors are willing to assess the effect of walking speed on foot inter-segment kinetics in the next study for additional insight into foot function during normal gait (*Dixon, Böhm & Döderlein, 2012*). Lastly, skin markers were used in this study as a non-invasive way to pursue segment movement and obtain kinematic data. However, skin markers oscillate and variability exists in identification of anatomical landmarks (*Karlsson & Tranberg, 1999*). To minimize the errors related to skin markers, a relatively small size markers (nine mm) compare to other protocols were used.
Further, an experienced laboratory technician were asked to attach the markers to all participants to avoid inter-tester variability.

## CONCLUSIONS

In summary, walking speed are shown to have significant effect on foot inter-segment motions, GRFs, and joint moments in healthy subjects as hypothesized. In the sagittal plane, limited HF/TB, and FF/HF motion of SW was likely to be compensated for increased HX/FF DF during stance phase. In the frontal and transverse planes, increased HF/TB EV and ER angles accompanied by decreased FF/HF SP in FW. Although the angle variation seems small, the subtle differences reflect complex motor control that adapts gaits in different walking speed. For kinetic parameters, increased peak vertical GRF in FW may affected by decreased stance duration time, the biomechanical mechanism under it may be smaller change in vertical COM height and increased leg stiffness under FW condition. Increased sagittal ankle moment is a consequence of increased walking speed. Walking speed changes accompanied with modulated sagittal plane ankle moments to alter the braking GRF during loading response. The propulsive GRF generated during push-off phase could modulate walking speed. Increased VALR and knee ADD moment in FW were assessed as marker of TB injury risks and knee joint loading. Kinetic parameters show a strong consistence with increases in walking speed, and was found mainly in differences in FW. The foot kinematic and kinetic data provided in this study from healthy subjects can be used as a reference set of normal gait parameters to distinguish abnormal gait patterns caused by different pathogenesis, and will provide a basis for clinicians when deciding on treatment for abnormal and pathological gait. On the other hand, measuring foot inter-segment kinematics and kinetics related to walking speed could provide guidance for intervention strategies aimed at improving walking gait performance.

### Funding
This work was supported by the National Natural Science Foundation of China (81772423), K. C. Wong Magna Fund in Ningbo University, and National Social Science Foundation of China (16BTY085). The funders had no role in study design, data collection and analysis, decision to publish, or preparation of the manuscript.

### Grant Disclosures
The following grant information was disclosed by the authors:
National Natural Science Foundation of China: 81772423.
K. C. Wong Magna Fund in Ningbo University.
National Social Science Foundation of China: 16BTY085.

### Competing Interests
The authors declare that they have no competing interests.

## Author Contributions

- Dong Sun conceived and designed the experiments, performed the experiments, analyzed the data, contributed reagents/materials/analysis tools, prepared figures and/or tables, authored or reviewed drafts of the paper.
- Gusztáv Fekete analyzed the data, contributed reagents/materials/analysis tools.
- Qichang Mei performed the experiments, analyzed the data, prepared figures and/or tables.
- Yaodong Gu conceived and designed the experiments, authored or reviewed drafts of the paper, approved the final draft.

## Human Ethics

The following information was supplied relating to ethical approvals (i.e., approving body and any reference numbers):

Ethics approval was obtained from the Ethics committee of Ningbo University (No. RAGH20170516).

## Data Availability

The raw data are provided in the Supplemental Files.

## Supplemental Information

Supplemental information for this article can be found online at http://dx.doi.org/10.7717/peerj.5517#supplemental-information.

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
