# Peer review of "The effect of walking speed on the foot inter-segment kinematics, ground reaction forces and lower limb joint moments"

_PeerJ, doi:10.7717/peerj.5517_

## Round 0.1 · original submission · Major Revisions

The reviewers have raised many questions that need to be addressed in a revised manuscript. Although novelty is not a requirement for publication in PeerJ, the presented study should be appropriately integrated in existing literature. The authors should also clarify the aim of the study, justify the number of participants, clarify how moments and angles were calculated, and discuss speed-dependent changes in intra-foot kinetics.

Reviewer 1 ·

Basic reporting

1.1 The paper generally uses clear and unambiguous English through; however, the paper should be further reviewed for grammar / correct use of English. A few notable examples: Lines 104, 161-162, and elsewhere: The present tense should be used.

1.2 Literature references need to be expanded further to fully cover the topic of intra-foot kinetics: A number of researchers have investigated intra-foot kinetics using multi-segment foot models. I have provided a few references below:

- PC Dixon, H Böhm, L Döderlein. Ankle and midfoot kinetics during normal gait: a multi-segment approach. Journal of biomechanics 45 (6), 1011-1016
- Bruening, D. A., Cooney, K. M. and Buczek, F. L. (2012). Analysis of a kinetic multi-segment foot model. Part I: Model repeatability and kinematic validity. Gait and Posture 35, 529–534 (listed in the reference list, but not described in the text)

Experimental design

The methods are generally described with sufficient detail & information to replicate; however, a few additional details could improve the manuscript:

2.1 Line 99: The authors discuss the use of a modified plug-in gait model. In which way was the model modified? These details are important for assessing this paper.

2.2 Line 109: “non-wanted effects” is vague and should be revised to allow the reader to understand the rational for the actions taken.

2.3 Line 140: It is unclear what the term “extrinsic moment data” refers to. If the authors are describing external moments, this term in preferred in the literature.

2.4 Knee and ankle moment calculations: Why were these computed via Keenan et al.? Is this approach different to the one implemented in the Vicon modeller? The authors should explain further their analysis choice.

2.5 Data filtering: Why was a single filter used for kinematic and kinetic data? Often settings are tuned for each data type.

2.6 Line 162 and elsewhere: One-way ANOVA tests do not “prove” that differences exist. The word proved used in this context is a little strong. The ANOVA simply suggests that you can are confident in rejecting the null hypothesis. This section and potentially others should be revised.

2.7 FF/TB angles: These angles are generated by the OFM modeller, but are rarely reported in the literature as they are difficult to interpret. The authors should justify the importance of assessing this measure.

Validity of the findings

The results appear valid based on the methods implemented. Statistical approach is sound.

Additional comments

This paper misses a great opportunity to assess the effect of speed on intra-foot kinetics. It is clear that adding these variables would represent a significant amount of extra work for the authors, but it would greatly improve the manuscript. At the very least, the authors should include in their limitations that these intra-foot kinetic parameters are important and could be investigated in future studies.

Reviewer 2 ·

Basic reporting

This is a well written article.
The research in this field is important, however the data reported in this article is not new.
The message (increasing rom with increasing speed) of the article had published before by various autors. Adding the ground reaction forces is something new. I would like to emphasie this more in this article.

Experimental design

I would like to see a more specific aim of this article with a more specific hypothesis. The goal of the article should include a statement about ground reaction forces with different speeds, while this is new.
I would like to know, why there were 10 subjects included.
Which foot was analysed
Why slow and fast speed was 25% higher and lower compared to normal speed
How angles were calculated. It is known that the angles are calculated by the anatomical postion of patients (sagittal flexion/extension, frontal inversion/eversion, transverse abduction/adduction). However some models, for example in van hoeve et al angles were calculated with the foot in line with the tibia resulting in (sagittal flexion/extension, frontal abduction/addution, transverse inverion/eversion).
36 markers were used?? Was there a static trial. Were there markers removed after static trials.
How is it possible that the speed was highly comparable between subjects, while they were walking at self selected speed?
Why was the stance presented and not stance and stride?
Why was the stance not divided in a loading phase, midstance and push-off phase.

Validity of the findings

How do the autors reflect on the findings that several parameters have lower ROM with higher speed. This is not in line with dubbeldam et al. 2010 and van Hoeve et al. 2017. They found higher ROM or equal ROM with higher speed. I understand that there will be some compensation in the foot however to autors did not find this in comparable research.
Please explain in the discussion.

Additional comments

Introduction:
Significant relevance have been found between walking speed and spatial-temporal
53 parameters, lower limbs kinematics, kinetics and electromyography activities in both healthy and 54 pathology adults (Andriacchi, Ogle and Galante, 1977; Chiu and Wang, 2007; Chung and Wang, PeerJ reviewing PDF | (2018:05:28076:0:0:CHECK 6 May 2018)
Manuscript to be reviewed 55 2010; Dubbeldam et al., 2010; Holden, Chou and Stanhope, 1997; Tulchin et al., 2009).

I feel this article should be citated while it is higly comparable.
Medicine (Baltimore). 2017 Sep;96(35):e7907. doi: 10.1097/MD.0000000000007907.
The effect of age and speed on foot and ankle kinematics assessed using a 4-segment foot model.
van Hoeve S1, Leenstra B, Willems P, Poeze M, Meijer K.

---

## Round 0.2 · accepted · Accept

The authors have adequately addressed the reviewer comments.

Reviewer 1 ·

Basic reporting

no comment

Experimental design

no comment

Validity of the findings

no comment

Additional comments

The authors have made sufficient changes to warrant publication